# Bromodomain Protein BRD4 Accelerates Glucocorticoid Dysregulation of Bone Mass and Marrow Adiposis by Modulating H3K9 and Foxp1

**DOI:** 10.3390/cells9061500

**Published:** 2020-06-19

**Authors:** Feng-Sheng Wang, Yu-Shan Chen, Jih-Yang Ko, Chung-Wen Kuo, Huei-Jing Ke, Chin-Kuei Hsieh, Shao-Yu Wang, Pei-Chen Kuo, Holger Jahr, Wei-Shiung Lian

**Affiliations:** 1Core Laboratory for Phenomics and Diagnostics, Kaohsiung Chang Gung Memorial Hospital, Kaohsiung 83301, Taiwan; wangfs@ms33.hinet.net (F.-S.W.); ggyy58720240@gmail.com (Y.-S.C.); bulakuo@gmail.com (C.-W.K.); maggie2258tw@gmail.com (H.-J.K.); jorno0329@gmail.com (C.-K.H.); vip690221@gmail.com (S.-Y.W.); bank66882007@gmail.com (P.-C.K.); 2Department of Medical Research, Kaohsiung Chang Gung Memorial Hospital, Kaohsiung 83301, Taiwan; 3Graduate Institute of Clinical Medical Sciences, College of Medicine, Chang Gung University, Kaohsiung 83301, Taiwan; 4Department of Orthopedic Surgery, Kaohsiung Chang Gung Memorial Hospital, Kaohsiung 83301, Taiwan; kojy@cgmh.org.tw; 5Department of Anatomy and Cell Biology, University Hospital RWTH Aachen, 52074 Aachen, Germany; 6Department of Orthopedic Surgery, Maastricht University Medical Center, 6229 HX Maastricht, The Netherlands

**Keywords:** BRD4, glucocorticoid, H3K9, acetylation, JQ-1, Foxp1

## Abstract

Glucocorticoid provokes bone mass loss and fatty marrow, accelerating osteoporosis development. Bromodomain protein BRD4, an acetyl–histone-binding chromatin reader, regulates stem cell and tissue homeostasis. We uncovered that glucocorticoid inhibited acetyl Lys-9 at the histone 3 (H3K9ac)-binding Runx2 promoter and decreased osteogenic differentiation, whereas bromodomain protein 4 (BRD4) and adipocyte formation were upregulated in bone-marrow mesenchymal progenitor cells. BRD4 knockdown improved H3K9ac occupation at the Runx2 promoter and osteogenesis, but attenuated glucocorticoid-mediated adipocyte formation together with the unaffected H3K9ac-binding PPARγ2 promoter. BRD4 regulated epigenome related to fatty acid metabolism and the forkhead box P1 (Foxp1) pathway, which occupied the PPARγ2 promoter to modulate glucocorticoid-induced adipocytic activity. In vivo, BRD4 inhibitor JQ-1 treatment mitigated methylprednisolone-induced suppression of bone mass, trabecular microstructure, mineral acquisition, and osteogenic differentiation. Foxp1 signaling, marrow fat, and adipocyte formation in glucocorticoid-treated skeleton were reversed upon JQ-1 treatment. Taken together, glucocorticoid-induced H3K9 hypoacetylation augmented BRD4 action to Foxp1, which steered mesenchymal progenitor cells toward adipocytes at the cost of osteogenic differentiation in osteoporotic skeletons. BRD4 inhibition slowed bone mass loss and marrow adiposity. Collective investigations convey a new epigenetic insight into acetyl histone reader BRD4 control of osteogenesis and adipogenesis in skeleton, and highlight the remedial effects of the BRD4 inhibitor on glucocorticoid-induced osteoporosis.

## 1. Introduction

Physiological glucocorticoid is indispensable in sustaining differentiation capacity of osteogenic cells, underpinning bone mineral accretion to maintain skeletal tissue health [1] and bone regeneration [2]. While therapeutic glucocorticoids are commonly prescribed to compromise immune disorders [3], the adverse effects of glucocorticoid on skeletal tissue, like low bone mineral density [4], sparse microstructure, and weak biomechanics [5], increase the risk of osteoporotic fracture [6]. Severe trabecular bone loss [7] and marrow fat overproduction [8] are notable histopathologic features of glucocorticoid-mediated osteoporotic skeleton. The development of mineral acquisition loss and fatty marrow hints that osteogenic and adipogenic programs in mesenchymal progenitor cells are dysregulated in the osteoporotic bone microenvironment.

Increasing evidence has revealed that epigenetic pathways modulate gene transcription without altering genomic sequences, regulating osteogenesis of mesenchymal stem cells, bone tissue metabolism, and osteoporosis development [9]. Of epigenetic modification reactions, histone acetylation maintains the interaction of histone assembly and DNA sequence, sustaining promoter activity for gene transcription [10]. For example, acetyltransferase p300/CBP-associated factor (PACF) hyperacetylation of lysine 9 at histone 3 (H3K9) increases bone morphogenetic protein gene expression, upregulating osteogenic differentiation of mesenchymal stem cells [11]. Histone deacetylase 4 (HDAC4) accelerates Runx2 deacetylation and bone mass loss in mice deficient in Ca^2+^ leak channel in the endoplasmic reticulum compartment [12]. HDAC3 knockdown up-regulates acetyl H4 enrichment in osteoprotegrin promoter in osteoprogenitor cells, enhancing osteoclastic resorption and bone remodeling [13]. Inhibition of HDAC1 improves histone architecture at the Runx2 promoter, preserving the differentiation capacity of pre-osteoblasts upon co-incubating with multiple myeloma cells [14]. α-ketoglutarate overproduction decreases H3K27 acetylation, hindering osteogenesis of mesenchymal stem cells deficient in superoxide dismutase [15]. We previously showed that H3K9 hypoacetylation, osteoblast loss, and marrow adipose over-development were present in glucocorticoid-induced osteoporotic bone tissue [16]. The molecular events by which glucocorticoid-mediated repression of H3K9 acetylation accelerated the shift of mesenchymal stem cells into adipocytes warranted investigation.

Bromodomain-containing protein 4 (BRD4) is a member of the bromodomain extra-terminal protein family, containing α-helix bromodomains able to interact with ε-acetyl-lysine residues of histones [17]. BRD4 recognizes acetyl histone actions, orchestrating transcription elongation factor and RNA polymerase II to control gene transcription [18]. Increasing reports reveal that BRD4 signaling regulates osteogenesis, chondrogenesis, and adipogenesis. Increased BRD4 enrichment in osteoblastic-specific enhancers is present during osteogenic differentiation of mesenchymal stem cells [19]. BRD4 inhibition attenuates hydrogen peroxide-inhibited survival and extracellular matrix in chondrocytes [20]. Loss of BRD4 function suppresses transcription of adipogenic transcription factors PPARγ and C/EBPα, decreasing the adipocyte formation of adipogenic progenitor cells [21]. Little is known about how the acetyl histone reader controls the osteogenic and adipogenic specification of mesenchymal progenitor cells in glucocorticoid-mediated osteoporotic bone microenvironments.

This study aimed to characterize what role BRD4 may play in the glucocorticoid excess-mediated dysregulation of osteogenic differentiation and adipocyte formation of osteoprogenitor cells and investigated whether BRD4 inhibition changed glucocorticoid-induced bone loss or fatty marrow formation.

## 2. Materials and Methods

### 2.1. Bone-Marrow Mesenchymal Stem Cell Cultures

GIBCO^®^-C57BL/6 immortalized murine bone-marrow mesenchymal stem cells (Thermo Fisher Scientific Inc., Waltham, MA, USA) were incubated in Dulbecco’s modified Eagle medium (DMEM; #11885; Gibco, Thermo Fisher Scientific Inc., Waltham, MA, USA) with 10% fetal bovine serum (Gibco, Thermo Fisher Scientific Inc., Waltham, MA, USA) till 70–80% confluence. Cell cultures within 5 passages were used for experiments.

### 2.2. In Vitro Osteogenesis and Adipogenesis Assay

A total of 10^5^ cells/well (24-well plates) cells were incubated in DMEM, 10% fetal bovine serum, 10 mM β-glycerophosphate, and 50 μg/mL ascorbic acid with vehicle or 1 μM dexamethasone (Sigma-Aldrich, St. Louis, MO, USA) for 18 days. Mineralized matrices were detected using the Von Kossa Stain Kit (Abcam, Cambridge, UK), according to the manufacturer’s instructions. Von Kossa-stained mineralized matrices in each field (12.5× magnification) were evaluated using an inverted microscope, as previously described [22,23]. For adipogenic differentiation, 10^5^ cells/well were incubated in DMEM, 10% fetal bovine serum, 10 μg/mL insulin with vehicle or 1 μM dexamethasone for 12 days. Cytoplasmic oil droplets were detected using the Nile Red Stain Kit (Abcam, Cambridge, UK) and evaluated using a fluorescence microscope. Adipocytes in each field (200× magnification) were counted. Quantification of mineralized matrices or adipocytes was performed on 3 fields in each well and 3 wells in each experiment using the image analysis software of the microscope.

### 2.3. RNA Interference and cDNA Transfection

Hdac4 sequence was cloned and ligated into pUSEamp(+) (Upstate Biotechnology, Lake Placid, NY, USA). BRD4 RNAi (#75135), Foxp1 RNAi (#172237), Hdac4 RNAi (#166634), and scramble control were obtained from Applied Biosystems Biotechnology. A total of 10^6^ cells/well (12-well plates) were transfected with 100 ng of RNAi, cDNA vector, or scramble control using Lipofectamine^TM^ 3000 Transfection Reagent (Invitrogen^TM^; Thermo Fischer Scientific, Waltham, MA, USA) and incubated in a 37 °C humidified incubator for 24 h. The transfected cells were incubated in osteogenic or adipogenic media for the designated time.

### 2.4. RT-PCR

Cells in each well were harvested and lysed to extract total RNA using TRI reagent^®^ (Sigma-Aldrich), as previously described [24]. An amount of 1 μg of total RNA was reversely transcribed using ReadScript^®^ Two-Step cDNA Synthesis Kits (Sigma-Aldrich, St. Louis, MO, USA). The ABI 7900 Detection System (Applied Biosystems, Waltham, MA, USA) and 2× TaqMan^®^ Universal PCR Master Mix, together with customized primers for PCR analysis of BRD4 (forward 5′-ACACCTGCA CCTACCAGACTC-3′; reverse, 5′-CATCGGCTACAGTCTAGGCC-3′), Runx2 (forward, 5′-CCAG CAGCACTCCATAT CTC-3′; reverse, 5′-CAGCGTCAACACCATCATTC-3′), OCN (forward, 5′-TAC AACTCACCAATC AT AAC-3′; reverse, 5′-ACCTTCATACTTCAACTACT-3′), PPARγ2 (forward: 5′-ACCATGGTTGACACAGAGATGCCA-3′; reverse: 5′-AGGAATGCGAGTGGTCTTCCATCA-3′), Foxp1 (forward, 5′-ATAGCTAACCTCGCTCGCCTA-3′; reverse, 5′-CTCTTCATACTACGATGC ATA-3′), and β-actin (forward, 5′-GACGGCCAGGTCATCACTAT-3′; reverse: 5′-CTTCTGCATCCT GTCAGCAA-3′) were used for computing threshold values (C*_t_*) of PCR amplification, according to the manufacturer’s manuals. Fold changes of mRNA expression were calculated using 2^−ΔΔC*_t_*^ (ΔC*_t_*, C*_t_* of gene—C*_t_* of housekeeping gene; ΔΔC*_t_*, C*_t_* of glucocorticoid group—C*_t_* of vehicle group), as previously described [24].

### 2.5. Immunoblotting

Hdac4, BRD4, Foxp1, H3K9ac, and actin levels in cell lysates and bone tissue lysates were detected using goat anti-mouse Hdac4 (aa 1–19; HDAC-144), rabbit monoclonal Foxp1 (aa 350–450; EPR4113), BRD4 (aa 1312–1362; EPR5150), H3K9ac (aa 1–100; EPR16988; ChIP grade), and mouse monoclonal actin (aa 1–100; mAbcam 8226) antibodies, which were all obtained from Abcam, Cambridge, UK. Protein bands of interest were visualized using Thermo Scientific™ SuperSignal™ Western Blotting Kits (Thermo Fisher Scientific Inc., Waltham, MA, USA) with horseradish peroxidase-conjugated IgG, luminol substrate and peroxide, according to the manufacturer’s instructions.

### 2.6. Immunofluorescence

BRD4 and Foxp1 immunofluorescence in cell cultures were investigated using BRD4 and Foxp1 antibodies together with Immunofluorescence Application Solutions Kits (Cell Signaling Technology, Danvers, MA, USA). In brief, formaldehyde-fixed cells were blocked using blocking buffer with 5% normal goat serum and 0.3% Triton™ X-100 for 60 min and followed by incubating with antibodies at 4 °C for 16 h. Specimens were incubated in anti-mouse IgG fragment (Alexa Fluor^®^ 488 conjugate) or anti-mouse IgG conjugated Alexa Fluor^®^ 675 and covered with Prolong^®^ Gold Antifade Reagent (Thermo Fisher Scientific Inc., Waltham, MA, USA). Cells in each field displaying nuclear BRD4 or Foxp1 immunoreaction were evaluated using the Olympus Laser Confocal Microscope (Olympus, Tokyo, Japan). Three fields in each well, and 3 wells in each experiment, were randomly selected for quantification.

### 2.7. Chromatin Immunoprecipitation (ChIP)-PCR

Nuclear lysates of 10^7^ cells were prepared using the Nuclear Extraction Kit (ab113474, Abcam, Cambridge, UK). Upon formaldehyde crosslinking and sonication-mediated DNA shearing, H3K9ac, BRD4, Foxp1, and IgG immunoprecipitates in nuclear extracts were prepared using EZ-Magna ChIP™ A/G Chromatin Immunoprecipitation Kits (Millipore, Temecula, CA, USA) along with specific antibodies, according to the manufacturer’s instructions. In brief, 1 μg of antibodies, IgG, and anti-RNA polymerase, together with protein G magnetic beads, were added to specimens and incubated at 4 °C with rotation for 16 h. Protein G-chromatin complexes were harvested using a magnetic separator and washed using Low Salt, High Salt, and LiCl Immune Wash Buffers. Specimens were mixed with Proteinase K in ChIP Elution Buffer and incubated at 62 °C for 2 h and 95 °C for 10 min. DNA was harvested and concentrated using spin columns. A total of 1 ng DNA was mixed with PCR mixtures and Cy3-conjugated primers for Runx2 (−942~+28 bp; ENSMUSG00000039153), PPARγ2 (−1996~−1731 bp; ENSMUSG00000000440), Foxp1 (−249~+32 bp; ENSMUSG00000030067) promoters and positive control GADPH promoter (ENSMUSG00000207654). The enrichment of H3K9ac, BRD4, Foxp1, and IgG in Runx2 and PPARγ2 promoter was expressed as % input DNA.

### 2.8. Chromatin Immunoprecipitation-Sequencing (ChIP-seq)

A total of 10^7^ bone-marrow mesenchymal stem cells were incubated in osteogenic medium with 1 μM dexamethasone and 0.1 μM JQ-1 for 24 h. H3K9ac immunoprecipitates in vehicle, dexamethasone, and JQ-1-treated cells were extracted and subjected to genome-wide sequencing using Illumina HiSeq4000 system (Illumina, Inc., San Diego, CA, USA). Quality control of >20 M reads, trimming reads <150 bp, mapping mouse genome and peak characterization were performed using CLC Genomics Workbench version 10 along with the Transcription Factor ChIP-seq analysis pipeline, as previously described [25]. DESeq software R package version 1.16.0 was used to verify fold changes of RPM. Heatmap of transcriptions start sites between upstream and downstream 5 kb were plotted using gplots R Package version 2.17.0 (https://CRAN.R-project.org/package=gplots). BAM files were deciphered using Integrative Genomics Viewer version 2.4.13 and followed by importing to SeqMock version1.42.0 (cutoff, 200 bp). Ontology of aligned genes were characterized using gene set enrichment analysis together with the KEGG database.

### 2.9. Glucocorticoid-Induced Bone Loss in Mice

All animal protocols and veterinary care were in compliance with animal welfare guidelines and approved (Affidavit No. 20141030701) by the Animal Use and Care Committee of Kaohsiung Chang Chung Memorial Hospital. Twelve-week-old male C57L/B6 mice were subcutaneously injected with vehicle (*n* = 6) and 10 mg/kg/day methylprednisolone (*n* = 6) for 4 consecutive weeks. In a subset of the experiment, methylprednisolone-treated mice were intraperitoneally injected with 250 µg/kg/day JQ-1 (*n* = 6) for 4 weeks. At 3 and 9 days before euthanasia, each mouse was intraperitoneally injected with 5 mg/kg calcein (Sigma-Aldrich, St. Louis, MO, USA). At the end of the experiment, animals were euthanatized using an overdose of anesthetics. Bone tissue were dissected for μCT and histomorphometry.

### 2.10. μCT Analysis

Radiography of bone microarchitecture was performed using the SkyScan 1176 μCT system (Bruker, Belgium). Three hundred radiograms with 9-μm voxel size of proximal tibiae were reconstructed using SKYSCAN^®^ CT-Analysis software, according to the manufacturer’s manuals. Bone mineral density of cortical bone and trabecular bone were quantified upon calibration of calcium hydroxyapatite phantom (Micro-CT HA; QRM GmbH, Moehrendorf, Germany). Trabecular thickness (Tb.Th, mm), trabecular number (Tb.N), trabecular separation (Tb.Sp), and structure model index (SMI) were measured automatically.

### 2.11. Ex Vivo Osteogenesis and Adipogenesis

After euthanasia, bone marrow in femurs and tibiae in vehicle, methylprednisolone, and JQ-1-treated mice was aspirated and mixed with RBC lysis buffer to remove red blood cells. Nucleated cells were incubated in DMEM with 10% fetal bovine serum for 24 h. After removing floating cells, adherent bone-marrow mesenchymal cells were harvested, as previously described [26]. A total of 10^5^ cells/well (24-well plates) were incubated in osteogenic medium for 18 days. Mineralized matrix formation was detected using von Kossa staining. In some experiments, 10^5^ cells/well (24-well plates) were grown in adipogenic medium for 12 days. Adipocytes were probed using Oil Red O staining. The mineralized matrix area positive for von Kossa stain and cells positive for Oil Red O stain in each field were counted, as described above.

### 2.12. Histomorphometry

Methyl acrylate-embedded tibiae blocks were prepared and micro-dissected into 7-μm sections, as previously described [24]. Trabecular and marrow adipose histology of each section was probed using the von Kossa Stain Kit and hematoxylin and eosin staining, respectively. Fluorescence calcein-labeled mineral deposition was evaluated by fluorescence microscopy. Trabecular bone volume (BV/TV, %), mineral acquisition rate (MAR, μm/day), and adipocyte number in 3 fields in each section and 3 sections in each mouse were calculated.

### 2.13. Statistical Analysis

All cell culture experiments were repeated 3 times. Differences of analysis among 3 groups were investigated using the ANOVA test and the Bonferroni post hoc test. *p* value <0.05 was considered statistically different.

## 3. Results

### 3.1. H3K9 Hypomethylation Dysregulated Osteogenic and Adipogenic Differentiation

First, we investigated whether H3K9 acetylation affected osteoblast differentiation or adipocyte formation in glucocorticoid-treated osteoprogenitor cells. Bone-marrow mesenchymal stem cells were incubated in osteogenic medium with and without 1 μM dexamethasone, an in vitro model mimicking glucocorticoid-induced bone loss. H3K9ac levels were decreased in the glucocorticoid-treated group, whereas Hdac4 was upregulated within 24 h upon treatment (Figure 1a). Hdac4 knockdown increased H3K9ac levels (Figure 1b) and attenuated glucocorticoid-induced loss of von Kossa-stained mineralized matrix formation 18 days after treatment (Figure 1c). Loss of Hdac4 function significantly reversed glucocorticoid-mediated inhibition of osteocalcin and osteogenic transcription factor Runx2 expression (Figure 1d) and H3K9ac binding Runx2 promoter 24 h after treatment, as evident from the chromatin immunoprecipitation (ChIP)-PCR analysis (Figure 1e). Osteogenic differentiation and H3K9ac-binding Runx2 promoter were significantly increased above baseline in Hdac4 RNAi-transfected cells, as compared to the scramble control group. Forced Hdac4 expression downregulated H3K9ac levels (Figure 1b), mineralized matrix deposition (Figure 1c), osteocalcin and Runx2 expression (Figure 1d), as well as significantly decreasing H3K9ac enrichment in the Runx2 promoter (Figure 1e), compared to the vector group. As cells grew in an adipogenic condition, glucocorticoid or Hdac4 overexpression significantly increased adipocyte formation, as evidenced in the fluorescent Nile red staining 12 days after treatment (Figure 1f), as well as upregulated adipogenic transcription factor PPARγ2 expression 24 h after treatment (Figure 1g). The glucocorticoid-induced upregulation of adipocytes and PPARγ2 expression were decreased in Hdac4 RNAi-transfected cells. Of note, for glucocorticoid, the gain or loss of Hdac4 function did not significantly change H3K9ac enrichment in PPARγ2 promoter 24 h after treatment (Figure 1h), indicating that an epigenetic reading pathway may regulate glucocorticoid-mediated PPARγ2 transcription.

### 3.2. BRD4-Regulated Osteogenic and Adipogenic Differentiation

BRD4 is known to read acetyl lysine of histone, altering chromatin architecture to determine stem cell fate [27] and adipogenesis [20]. We examined whether the molecule was involved in the glucocorticoid-mediated adipocyte formation of mesenchymal stem cells. BRD4 mRNA expression and protein levels were significantly increased upon glucocorticoid treatment (Figure 2a). Glucocorticoid-treated cells showed strong BRD4 immunofluorescence in the nuclear compartment 24 h after treatment (Figure 2b). While H3K9ac levels were downregulated upon glucocorticoid treatment, BRD4 levels were increased in H3K9ac immunoprecipitates (Figure 2c). BRD4 knockdown (Figure 2d) significantly mitigated glucocorticoid-mediated suppression of the H3K9ac-binding Runx2 promoter (Figure 2e), improving osteocalcin and Runx2 expression (Figure 2f), and mineralized matrix accumulation (Figure 2g). Loss of BRD4 function repressed glucocorticoid-mediated adipocyte formation (Figure 2h) and decreased PPARγ2 expression (Figure 2i); however, H3K9ac enrichment in the PPARγ2 promoter was unaffected in glucocorticoid-treated or BRD4 knockdown cells (Figure 2j).

### 3.3. BRD4 Inhibition Changed H3K9ac Enrichment at Coding Region of Lipid Regulators

We adopted genome-wide chromatin immunoprecipitation sequencing (ChIP-seq) to verify how BRD4 signaling regulated PPARγ2 transcription during glucocorticoid-mediated adipocyte formation. To this end, cells were incubated in vehicle, glucocorticoid or BRD4 inhibitor JQ-1, which reduced glucocorticoid-induced upregulation of BRD4 levels (Figure 3a). Upon treatment for 24 h, H3K9ac in cell cultures was immunoprecipiated and subjected to ChIP-seq. H3K9ac occupancy within 5 kb upstream and downstream of the transcription start site was decreased in glucocorticoid-treated cells, whereas it was reversed in JQ-1-treated cells, as evidenced in the heatmap (Figure 3b) and coverage plots (Figure 3c). Thirty-four H3K9ac-enriched binding sites were significantly altered in three groups (Figure 3d). In total, 11.31% and 14.07% of them distributed in promoter and coding regions (Figure 3e), respectively. Ontology analysis revealed that BRD4 inhibition changed lipogenesis and fatty acid metabolism, like long-chain fatty acid metabolism, membrane lipid biosynthesis, and lipid transport, etc. (Figure 3f). BRD4 significantly affected H3K9 enrichment in the binding domain of seven transcription factors, including metal regulatory transcription factor 1 (Mtf1), transcription factor 3 (Tcf3), forkhead box L1 (Foxl1), Forkhead box P1 (Foxp1), E47-like ETS transcription factor 3 (Elf3), high mobility group protein 1 (HMG-1), and myocyte enhancer factor 2C (Mef2c) (Figure 3g). Of the transcription factors, Foxp1 is reported to regulate adipogenic differentiation of adipogenic progenitor cells and energy metabolism of adipose tissue [28] and was selected for succeeding experiments.

### 3.4. Foxp1 Mediated Glucocorticoid Upregulation of Adipocyte Formation

Consistent with the ChIP-seq analyses, glucocorticoid significantly increased H3K9ac (Figure 4a) and BRD4 (Figure 4b) occupation at Foxp1 promoter, upregulating Foxp1 mRNA expression (Figure 4c) and protein levels (Figure 4d). BRD4 knockdown significantly decreased BRD4 and H3K9ac-binding Foxp1 promoter below baseline, as well as repressing glucocorticoid-mediated Foxp1 expression. Nuclei in glucocorticoid-treated mesenchymal stem cells exhibited strong Foxp1 immunofluorescence, whereas the immunoreaction was lessened in BRD4 knockdown cells (Figure 4e).

We investigated whether Foxp1 affected the osteogenic or adipogenic differentiation capacity of glucocorticoid-treated mesenchymal progenitor cells. Foxp1 interference significantly repressed glucocorticoid-induced Foxp1 mRNA expression and protein levels (Figure 5a), as well as decreasing the transcription factor binding PPARγ2 promoter (Figure 5b). Consistently, loss of Foxp1 function counteracted glucocorticoid-mediated PPARγ2 expression (Figure 5c) and adipocyte formation (Figure 5d). Glucocorticoid or Foxp1 knockdown did not significantly change the Foxp1-binding Runx2 promoter (Figure 5e). Foxp1 interference did not significantly affect glucocorticoid-induced Runx2 loss (Figure 5f) and mineralized matrix underproduction (Figure 5g).

### 3.5. BRD4 Inhibitor Treatment Slowed Glucocorticoid-Induced Bone Loss

Given that BRD4 inhibition attenuated the glucocorticoid-induced loss of the osteogenic differentiation capacity of mesenchymal stem cells, we tested whether BRD4 inhibitor administration affected glucocorticoid-induced bone mass loss or fatty marrow. Vehicle and 10 mg/kg/day methylprednisolone-treated mice were given 250 μg/kg/day JQ-1 for 4 consecutive weeks (Figure 6a). Glucocorticoid-treated skeletal tissue showed sparse trabecular bone microstructure as compared to the vehicle group, whereas mild bone loss was present in JQ-1-treated mice (Figure 6b). Cortical BMD (Figure 6c), trabecular BMD (Figure 6d), trabecular number (Tb.N; Figure 6e), and trabecular thickness (Tb.Th; Figure 6f) were significantly decreased in glucocorticoid-treated skeleton. Structure model index (SMI) of trabecular was significantly increased in glucocorticoid-treated bone tissue (Figure 6g), whereas trabecular separation (Tb.Sp) (Figure 6h) was comparable to vehicle-treated skeleton. Administration with JQ-1 significantly delayed glucocorticoid-mediated loss of bone mineral density, Tb.N, and Tb.Th. Tb.Sp, rather than SMI, was significantly improved in JQ-1-treated skeleton.

The JQ-1 improvement of bone mass prompted us to examine whether the treatment changed glucocorticoid-mediated BRD4, Foxp1 signaling, osteogenic or adipocytic activity. BRD4 and Foxp1 levels were increased, whereas H3K9ac levels were down-regulated in the methylprednisolone-treated group. These signals were compromised in the JQ-1-treated group (Figure 7a). Glucocorticoid-treated skeletons showed trabecular loss histopathology (Figure 7b) and poor mineral accumulation, as evidenced in the fluorescence calcein deposition (Figure 7c) together with significant reductions in BV/TV and mineral acquisition rate (MAR). Marrow adipose overdevelopment (Figure 7d) along with increased adipocyte number (Figure 7e) were present in the glucocorticoid-treated group. JQ-1 treatment reversed the glucocorticoid-mediated downregulation of trabecular histology and mineral accretion (Figure 7b,c), as well as improving marrow fat overproduction (Figure 7d,e). Oil Red O staining revealed that adipocyte formation of primary bone-marrow mesenchymal cells was significantly increased in glucocorticoid-treated skeleton, whereas JQ-1 treatment significantly mitigated the effect (Figure 7f). Von Kossa staining showed mineralized matrix underproduction in primary bone-marrow stromal cells from methylprednisolone-treated bone. JQ-1 treatment significantly improved osteogenic differentiation capacity (Figure 7g).

## 4. Discussion

Chronic glucocorticoid medication results in poor bone mineralization together with marrow fat overproduction in skeletal microenvironments, putting bone tissue at an incredibly high risk of osteoporosis [29]. Adipocyte formation at the expense of the osteogenic differentiation capacity of mesenchymal stem cells accelerates the development of glucocorticoid-induced bone loss and marrow lipid accumulation [30,31]. Increasing evidence reveals that epigenetic pathways, like DNA methylation [32], microRNA [33], and post-translational modification of histones [34], control lineage commitment programs of mesenchymal stem cells in pathological contexts. While histone acetylation is shown to preserve osteogenic differentiation of bone-marrow mesenchymal stem cells isolated from estrogen deficiency [35] or unloading-induced osteoporotic skeletal tissue [36], still little is understood about how acetyl histone regulates marrow adiposity during glucocorticoid-induced osteoporosis. This study uncovered that BRD4 interacted with H3K9ac, increasing Foxp1 action to drive mesenchymal stem cells into adipocytic cells in glucocorticoid-mediated osteoporotic bone. Collective evidence conveys a new epigenetic insight into glucocorticoid-induced bone loss and marrow adipocyte overgrowth and highlights the remedial effect of BRD4 inhibitor to osteoporosis.

H3K9 hypoacetylation was present in the glucocorticoid-mediated dysregulation of osteogenic and adipogenic differentiation of mesenchymal stem cells. Analyses were in agreement with other groups’ studies revealing that H3K9ac is involved in osteogenic differentiation of bone-marrow stromal cells [11] and glucocorticoid-induced loss of osteoblastic activity [37]. H3K9ac action to marrow adipocyte formation within osteoporotic skeleton remains uncertain. For example, H3K9ac levels are decreased during adipogenic differentiation of human amniotic fluid mesenchymal stem cells [38], whereas the acetyl histone level is unchanged in the adipocyte overdevelopment of bone-marrow stromal cells from glucocorticoid-mediated osteoporotic rats [39]. Jang et al. report that the acetyl histone is increased in the adipocyte formation of murine 3T3-L1 adipogenic progenitor cells [40]. This study uncovered that H3K9ac-binding PPARγ2 promoter was unaffected, whereas PPARγ2 transcription and adipocytic activity were upregulated in glucocorticoid-treated cells. We speculated that different extracellular stresses and mesenchymal progenitor cells originated from different tissues may result in different acetylation states of H3K9. The investigations also prompted us to hypothesize that an acetylated histone reading mechanism may influence these events.

Bromodomain protein BRD4 consists of acetyl lysine binding domains, which interact with acetylated histones to regulate chromatin remodeling [41]. Accumulating evidence reveals that BRD4 controls the lineage specification of stem cells, like mesendoderm specification of pluripotent stem cells [42], macrophage differentiation of hematopoietic stem cells [43], and osteogenic differentiation of mesenchymal stem cells [18]. The biological function of BRD4 signaling to osteogenic differentiation and adipocyte formation in glucocorticoid-mediated osteoporotic bone warranted investigations. This study is the first indication revealing that BRD4 directed bone-marrow mesenchymal stem cells into adipocytes at the expense of osteoblast differentiation, as knockdown of the acetyl histone reader protein attenuated glucocorticoid-induced adipocyte overgrowth, reversing mineralization matrix production. Increasing studies show that BRD4 inhibitor treatment decreases transcription of adipocyte marker lipoprotein lipase in 3T3-L1 adipogenic progenitor cells [44] and downregulates lipid accumulation in hepatic tissue in fructose-fed mice [45]. Of interest, BRD4 signaling loss changed H3K9a occupation at genes related to fatty acid metabolism and lipogenesis, as evidenced in the ChIP-seq analyses, which are consistent with the investigations of decreased adipocyte formation in BRD4 knockdown cells. Profound findings shed new light onto the effect of acetyl-histone reader on glucocorticoid-induced osteogenesis repression and adipocyte overdevelopment.

It is noteworthy that adipocyte formation was increased upon glucocorticoid treatment, whereas H3K9ac occupancy in the PPARγ2 promoter was unchanged either in glucocorticoid-treated or BRD4 knockdown cells. The investigations indicated that BRD4 may coordinate H3K9ac-mediated transcription factors to alter PPARγ2 transcription. Increasing reports have shown that the loss of BRD4 function recruits bio-active regulators, which facilitate adipocyte formation of 3T3-L1 cells [46] and TNF-α-treated human Simpson-Golabi-Behmel syndrome adipocytes [47]. In this study, the BRD4 alteration of H3K9ac binding 7 transcription factors explained the complex nature of epigenetic pathways in the development of glucocorticoid-mediated adipocyte formation. Foxp1 signaling appeared to, at least in part, regulate glucocorticoid-mediated PPARγ2 transcription as BRD4 or Foxp1 knockdown reduced Foxp1 occupation at the PPARγ2 promoter, ameliorating glucocorticoid-induced oil droplet overproduction. Analyses were in agreement with a study showing that Foxp1 intensively influences PPARγ2 transcription to regulate adipocyte function [28]. This study revealed that Foxp1 was required to accelerate the shift of glucocorticoid-treated mesenchymal stem cells into adipocytes at the expense of osteoblast differentiation. The investigations of unaffected Foxp1 occupation at the Runx2 promoter in glucocorticoid-treated or Foxp1 RNAi-transfected cells further explained the adipogenesis-promoting role Foxp1 signaling did play in mesenchymal stem cells. We do not exclude the possibility that other BRD4-regulated transcription factors, like HMG-1 or MEF2C [48,49,50], may affect the osteogenic or adipogenic differentiation capacity of glucocorticoid-treated mesenchymal stem cells. Robust findings revealed a new molecular mechanism underlying BRD4 control of lineage specification programs of mesenchymal progenitor cells in a glucocorticoid-mediated osteoporotic condition.

Consistent with the in vitro analyses showing that BRD4 loss preserved osteogenesis, administration with JQ-1 reduced Foxp1 levels and slowed the development of a plethora of osteoporosis signs, like poor bone mass, microarchitecture, decreased mineral acquisition, and low mineralized matrix production of bone-marrow mesenchymal cells in methylprednisolone-administered mice. Pharmacological inhibition of BRD4 is shown to inhibit osteoclast differentiation, preventing bone tissue from osteoporosis in ovariectomized mice [51]. Mice lacking Foxp1 show fatty marrow together with mesenchymal stem cell senescence and low bone mass [52]. Mice deficient in Foxp1 in perichondrium display defective osteogenesis phenotypes with premature osteoblast differentiation of mesenchymal progenitor cells [53]. Investigations of JQ-1 mitigation of marrow adiposity histopathology and adipocyte formation of bone-marrow stromal cells consolidated the evidence that BRD4 signaling loss drove glucocorticoid-treated mesenchymal progenitor cells away from adipocytes.

Taken together, H3K9ac together with BRD4 were required to maintain the Runx2 transcription of osteogenic differentiation of mesenchymal stem cells (Figure 8a). Glucocorticoid excess increased BRD4 actions, upregulating Foxp1 transcription to enhance PPARγ2 signaling, which accelerated the differentiation of mesenchymal stem cells into adipocytes (Figure 8b). This study offers an emerging BRD4-mediated epigenetic pathway, which determines mesenchymal stem cell fate in glucocorticoid-induced osteoporotic skeleton and fatty marrow, and highlights the perspective of BRD4 inhibitor protection against glucocorticoid overmedication-mediated bone disorders.

## Figures and Tables

**Figure 1 cells-09-01500-f001:**
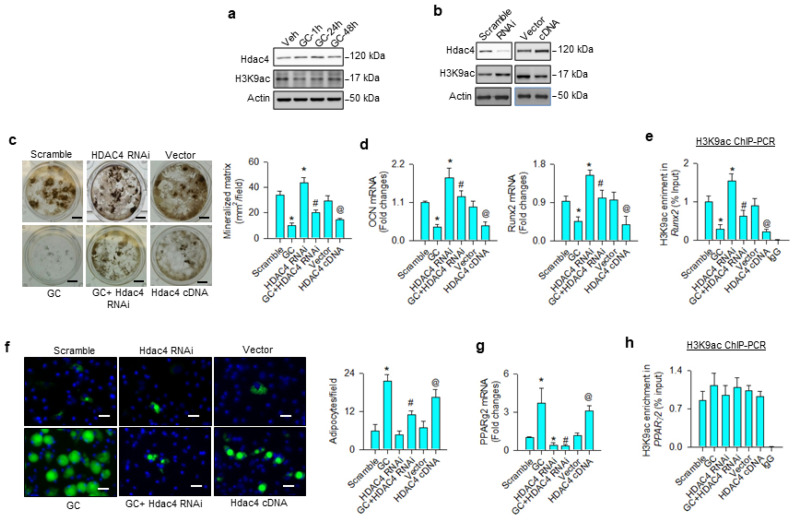
Analyses of H3K9ac action to glucocorticoid modulation of osteogenesis and adipogenesis of bone-marrow mesenchymal stem cells. Glucocorticoid increased Hdac4 levels, but reduced H3K9ac signaling (**a**). Hdac4 interference increased H3K9ac levels (**b**), attenuating glucocorticoid-induced repression of mineralized matrix formation (scale bar, 5 mm) (**c**), osteocalcin and Runx2 expression (**d**), as well as reversed H3K9ac enrichment in Runx2 promoter (**e**). Hdac4 knockdown alleviated glucocorticoid-mediated adipocyte formation (scale bar, 8 μm) (**f**) and PPARγ2 expression (**g**). Loss of Hdac4 function did not significantly change H3K9ac occupancy in the PPARγ2 promoter (**h**). Forced HDAC4 expression decreased H3K9ac levels, binding the Runx2 promoter and osteogenic differentiation, whereas adipocyte formation was increased. Mineralized matrix, adipocyte, mRNA expression, and H3K9ac enrichment in the promoter were probed using von Kossa staining, Nile Red fluorescence staining, qRT-PCR, and ChIP-PCR, respectively. Data are expressed as mean ± SEM calculated from 3 experiments. Asterisks * indicate significant differences from the scramble group, hashtag # indicates significant differences from the GC group (*p* < 0.05), and @ indicates significant differences from the vector group. GC: glucocorticoid; Veh: vehicle.

**Figure 2 cells-09-01500-f002:**
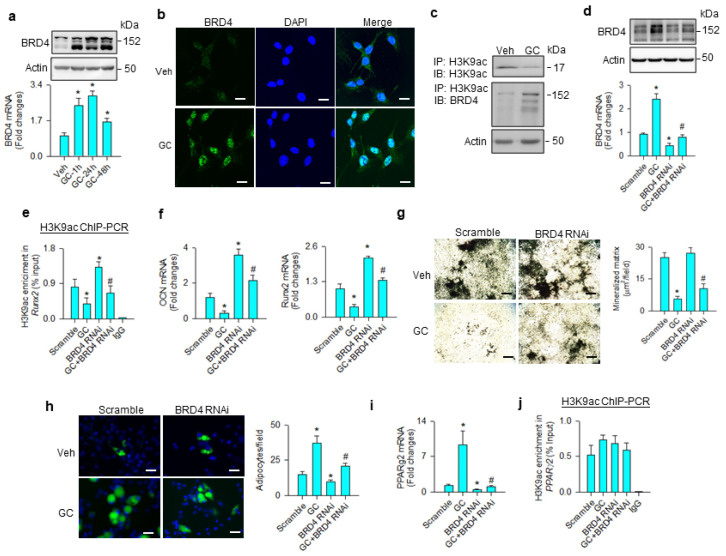
Analyses of BRD4 action to glucocorticoid-mediated osteogenic differentiation and adipocyte formation of bone-marrow mesenchymal stem cells. Glucocorticoid increased BRD4 mRNA expression and protein levels (**a**). Glucocorticoid-treated cells showed strong BRD4 immunofluorescence (scale bar, 8 μm) (**b**). H3K9ac levels were decreased upon glucocorticoid treatment, whereas BRD4 was increased in the H3K9ac immunocomplex (**c**). BRD4 interference decreased BRD4 mRNA expression and protein levels (**d**) and improved glucocorticoid-induced repression of H3K9ac occupation at the Runx2 promoter (**e**), osteocalcin and Runx2 mRNA expression (**f**), and mineralized matrix production (**g**); scale bar, 40 μm. BRD4 knockdown attenuated adipocyte formation (scale bar, 8 μm) (**h**) and PPARγ2 mRNA expression (**i**) rather than H3K9ac enrichment in the PPARγ2 promoter (**j**) in glucocorticoid-treated cells. Mineralized matrix, adipocyte, mRNA expression, and H3K9ac enrichment in the promoter were probed using von Kossa staining, Nile Red fluorescence staining, qRT-PCR, and ChIP-PCR, respectively. Data are expressed as mean ± SEM calculated from 3 experiments. Asterisks * indicate significant differences from the scramble group and hashtag # indicates significant differences from the GC group (*p* < 0.05). GC: glucocorticoid; Veh: vehicle.

**Figure 3 cells-09-01500-f003:**
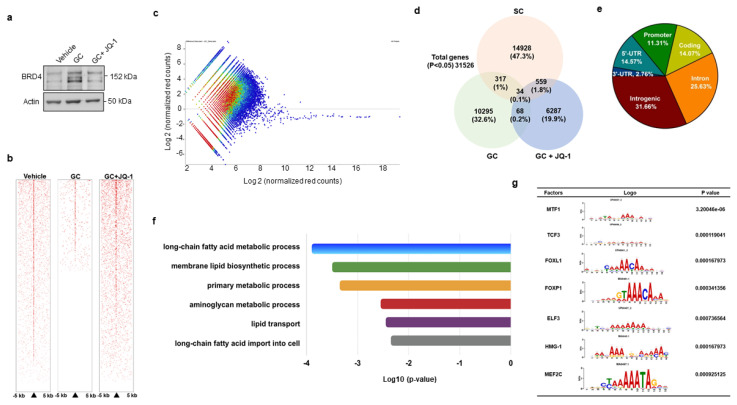
ChIP-seq analyses of H3K9ac-binding genes in vehicle-, glucocorticoid-, and JQ-1-treated mesenchymal stem cells. JQ-1 treatment attenuated glucocorticoid-induced BRD4 levels (**a**). Heatmaps of ChIP-seq analyses showing H3K9ac occupancy in DNA within 5 kb upstream and downstream of transcriptional start site (**b**). Scatter plots of H3K9ac ChIP-seq signals distribution in Veh, GC, GC + JQ-1 groups (**c**). Venn diagram displaying the overlap of H3K9ac-enriched genes (**d**). Relative distribution of H3K9ac occupancy across genomic regions (**e**). Gene set enrichment analysis of H3K9ac marks showed significant changes in fatty acid and lipid metabolism in the Vehicle, GC, and GC + JQ-1 groups (**f**). The H3K9ac-enriched motif of transcription factors identified in the vehicle, GC, and GC + JQ-1 groups (**g**). ChIP-seq of H3K9ac-enriched epigenome was performed using Illumina HiSeq4000 system. Experiments were performed three times. Veh: vehicle; GC: glucocorticoid.

**Figure 4 cells-09-01500-f004:**
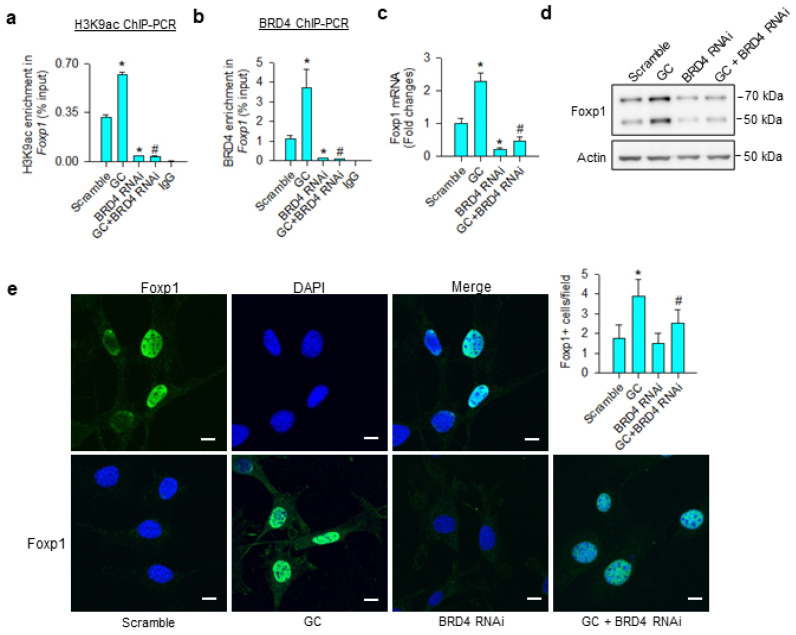
Analyses of glucocorticoid and BRD4 actions to Foxp1 expression in bone-marrow mesenchymal stem cells. BRD4 knockdown attenuated glucocorticoid-mediated upregulation of H3K9ac enrichment (**a**) and BRD4 occupation (**b**) at Foxp1 promoter along with significant reductions in Foxp1 mRNA expression (**c**) and protein levels (**d**). Glucocorticoid-treated cells showed strong Foxp1 immunofluorescence in the nuclear compartment, whereas the immunoreaction was attenuated in BRD4 RNAi-transfected cells (**e**); scale bar, 8 μm. H3K9ac and BRD4 enrichment in the promoter was probed using ChIP-PCR; Foxp1 expression was quantified using qRT-PCR, immunoblotting, and immunofluorescence staining. Data are expressed as mean ± SEM calculated from 3 experiments. Asterisks * indicate significant differences from the scramble group and hashtag # indicates significant differences from the GC group (*p* < 0.05). GC: glucocorticoid.

**Figure 5 cells-09-01500-f005:**
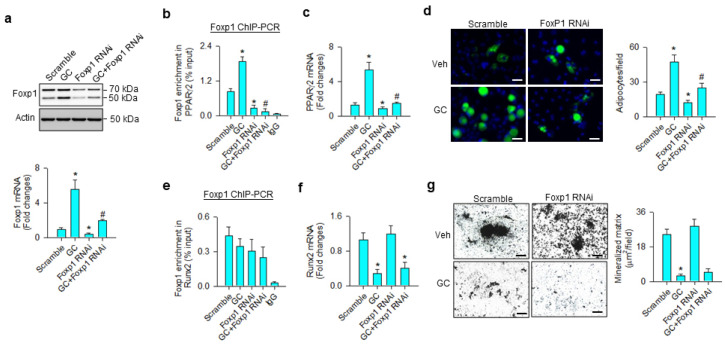
Analyses of Foxp1 action to glucocorticoid modulation of osteogenic and adipogenic differentiation. Foxp1 interference decreased Foxp1 mRNA expression and protein levels (**a**) and attenuated glucocorticoid-mediated upregulation of Foxp1 enrichment in the PPARγ2 promoter (**b**), PPARγ2 expression (**c**), and adipocyte formation (**d**); scale bar, 8 μm. Foxp1 knockdown did not significantly affect Foxp1 enrichment in the Runx2 promoter (**e**). Loss of Foxp1 function did not significantly change glucocorticoid-induced Runx2 loss (**f**) or mineralized matrix underproduction (**g**); scale bar, 40 μm. Foxp1 enrichment in promoters, mineralized matrix, and adipocyte formation were probed using ChIP-PCR, von Kossa staining, and Nile red fluorescence staining. Data are expressed mean ± SEM calculated from 3 experiments. Asterisks * indicate significant differences from the scramble group and hashtag # indicates significant differences from the GC group (*p* < 0.05). GC: glucocorticoid; Veh: vehicle.

**Figure 6 cells-09-01500-f006:**
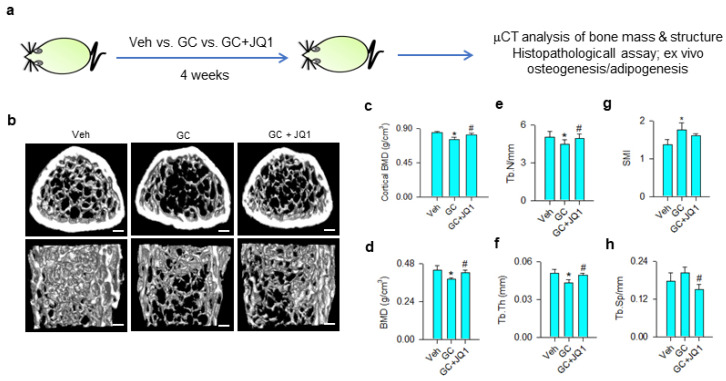
μCT analyses of bone mass and microstructure of glucocorticoid- and JQ-1-treated mice. Schematic drawing for administration with glucocorticoid and JQ-1 (**a**). Glucocorticoid-treated bone tissue showed poor trabecular bone microarchitecture, whereas mild bone loss was present in the GC + JQ-1 group (**b**); Scale bar, 5 mm. JQ-1 treatment attenuated glucocorticoid-mediated loss of cortical BMD (**c**), trabecular BMD (**d**), Tb.N (**e**), and Tb.Th (**f**). SMI were increased in the glucocorticoid-treated group (**g**), whereas Tb.Sp were decreased in the GC + JQ-1 group (**h**). Bone mass and microarchitecture were probed using the Skyscan 1176 μCT system together with SKYSCAN^®^ CT-Analysis software. Data are expressed mean ± SEM calculated from 6 mice. Asterisks * indicate significant differences from the vehicle group and hashtag # indicates significant differences from the GC group (*p* < 0.05). GC: glucocorticoid; Veh: vehicle.

**Figure 7 cells-09-01500-f007:**
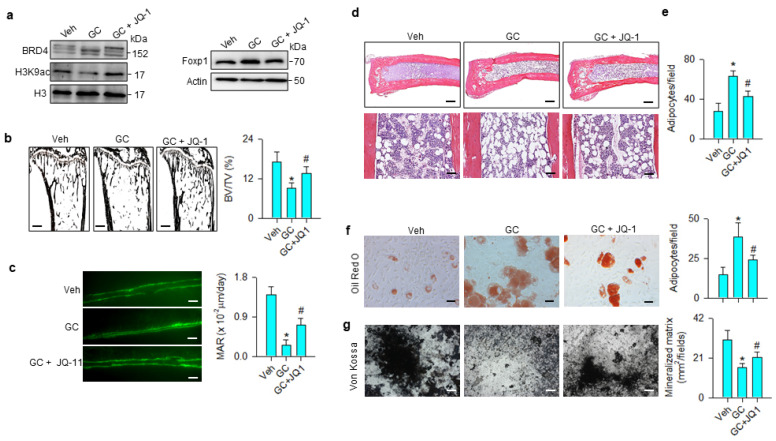
Effects of JQ-1 and methylprednisolone treatment on Foxp1 signaling, bone histology, and ex vivo osteogenesis and adipogenesis. JQ-1 treatment attenuated glucocorticoid-mediated upregulation of BRD4 and Foxp1 levels and improved H3K9ac levels (**a**). Glucocorticoid-induced loss of trabecular bone histology (scale bar, 120 μm) (**b**) and fluorescent calcein-labeled mineral acquisition (scale bar, 30 μm) (**c**), and marrow adiposity (Upper scale bar, 25 μm; lower scale bar, 8 μm) (**d**,**e**) were improved in the GC + JQ-1 group. JQ-1 treatment mitigated glucocorticoid-augmented adipocyte formation (scale bar, 8 μm) (**f**) and improved mineralized matrix formation (scale bar, 10 μm) (**g**) of primary bone-marrow mesenchymal cells. Trabecular histology, mineralization and adipocyte formation of bone-marrow stromal cells were probed using von Kossa staining, calcein labeling, and Oil Red O staining, respectively. Data are expressed as mean ± SEM calculated from 6 mice. Asterisks * indicate significant differences from the vehicle group and hashtag # indicates significant differences from the GC group (*p* < 0.05). GC: glucocorticoid; Veh: vehicle.

**Figure 8 cells-09-01500-f008:**
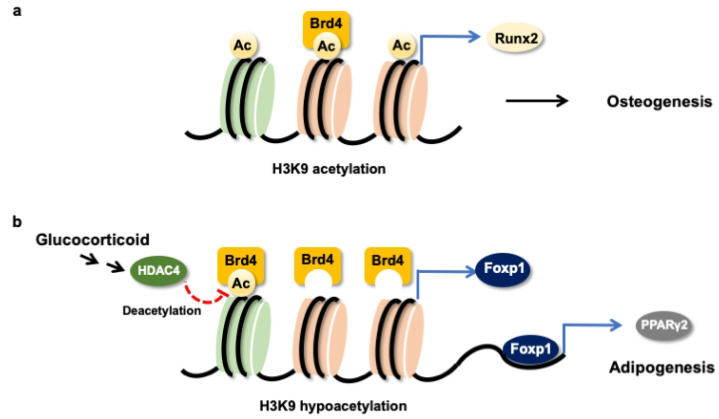
Schematic drawing of BRD4 control dysregulation of osteogenic differentiation and adipocyte formation of bone-marrow osteoprogenitor cells. BRD4 and H3K9ac are required in the Runx2 transcription in the osteogenic differentiation of mesenchymal stem cells (**a**). Glucocorticoid increases HDAC4 and BRD4 actions to Foxp1 transcription, which upregulates PPARγ2 signaling to drive mesenchymal stem cells into adipocytes (**b**).

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
