# Peer review of "Bromodomain Protein BRD4 Accelerates Glucocorticoid Dysregulation of Bone Mass and Marrow Adiposis by Modulating H3K9 and Foxp1"

_cells, 2020, doi:10.3390/cells9061500_

Round 1

Reviewer 1 Report

SUMMARY

This study examines a pathway by which glucocorticoids (GCs) promote BMSC differentiation into adipocytes rather than osteocytes. The authors show that in vitro GC treatment of BMSCs transiently increases H3K9 acetylation, and results in BRD4, Foxp1, and PPARg2 expression and adipocyte differentiation. This is a nice set of experiments, but some data that contradict the model appear to be ignored. A more cautious interpretation of some of the data would make the paper more convincing.

MAJOR COMMENTS

Fig 1: It is unclear if all the correct controls were done, since siRNA transfection and plasmid transfection will have different effects on cells (as clearly demonstrated by the difference in H3K9ac between scrambled siRNA and empty vector plasmid, which is more dramatic than any of the experimental treatments of interest). For a clear comparison, the knockdown experiment should have 1: scramble + vehicle, 2: scramble + dexamethasone, 3: siRNA + vehicle, 4: siRNA + dexamethasone. Simiarly, the overexpression experiment should have 1: vector + vehicle, 2: vector + dexamethasone, 3: cDNA + vehicle, 4: cDNA + dexamethasone. Without these fundamental controls, and especially with the clear difference between scramble RNAi and empty vector in H3K9ac, these data aren't really interpretable.

PLA experiment: It seems to me that there are several issues with the PLA data. First, nearly all the red H3K9ac-BRD4 PLA signal is in the cytosol, rather than the nucleus. Doesn't this suggest that the signal is all nonspecific, as the histones should all be in the nucleus? Second, the signal is too diffuse to quantitate spot counts as stated in the methods. Third, if there is nuclear signal, it is higher in vehicle- rather than GC-treated cells, thus suggesting that any BRD4-H3K9ac interaction is actually decreased upon GC treatment. Thus, the PLA assay either didn't work or it disagrees with the other data on BRD4 function.

ChIP-seq: What size of chromatin fragments were obtained? Also, I believe that ENCODE's recommendation for ChIP-seq experiments is to obtain >20 million uniquely mapping reads per sample (Landt et al., Genome Research, 2012), was this obtained? It is not completely clear in the methods section, and figure 3 indicates a total of 30,000 reads which would be problematic. Furthermore, the workflow used a "Transcription Factor ChIP-seq analysis pipeline" but this experiment was performed on histone protein, not a transcription factor (requiring different analysis, again according to the above ENCODE recommendations).

Fig 3: It would be very interesting to then see the H3K9ac ChIP-seq data for control samples in Fig 3. In 3A, for example, do the Control sample peaks look like GC+JQ1? And in Fig 3C, how much does the control overlap with the other two samples? I think the control samples really need to be shown to make the ChIP-seq experiment interpretable. Furthermore, the genes in D and E, which geneset is used for this analysis? Genes upregulated with GC treatment but downregulated in GC+JQ1 treatment?

Foxp1 data: The microscopy images in Figure 5e indicate that Foxp1 RNAi treatment dramatically increases adipocyte generation, rather than decreasing it. More generally, the authors present BMSC differentiation into osteocyte or adipocyte as an either/or decision, but the loss of Foxp1 has no effect on mineralization and Runx2. How do the authors explain this?

MINOR COMMENTS

Grammar and language editing would help with readability.

The figure quality should be improved - some of the text is difficult to read.

Figure captions should explain the method and experiment, but are probably not the correct place to interpret the results.

Fig 1: GC effects on HDAC4 and H3K9ac are time dependent (altered at 1, 24 h, back to normal at 48 h) so treatment times in other experiments should be explicitly stated. This also relates to interpretation, for example in L.233-234 the authors state that "HDAC4 signaling was augmented throughout the study period" but actually, this data only shows HDAC4 protein level, and also it is back to baseline at 48 h GC treatment. This should be corrected.

Fig 5B, 5C - if these are ChIP-PCR experiments, maybe this should be labeled in the figure? Just to be consistent with the previous figures.

Reviewer 2 Report

Bromodomain Protein BRD4 Accelerates Glucocorticoid Dysregulation of Bone Mass and Marrow Adipose by Modulating H3K9 and Foxp1 by Feng-Sheng Wang et al. is a generally comprehensive study. 

   That said, the work shows that glucocorticoids inhibit H3K9ac binding to the Runx2 promoter with decreased osteogenic differentiation, while BRD4 and adipocyte formation are upregulated in bone-marrow MSC. There is supporting data for BRD4 effects on Runx2 and osteogenesis, with negative effects on glucocorticoid-mediated adipocyte formation. Mechanistic data is shown regarding the Foxp1 pathway. The in vivo data are helpful; a BRD4 inhibitor opposed prednisone suppression of bone mass and related parameters. Reliance on this inhibitor only, and relatively small in vivo effects are a problem. Overall, the authors conclude that glucocorticoid-induced H3K9 hypoacetylation augments BRD4 action to Foxp1, directing MSC to fat differentiation relative to bone. The conclusions are overstated relative to the in vitro findings, however. 

  1. The work needs editing for grammar, including in the abstract. Overall, the presentation is telegraphic and revision for clarity would be useful. 
  2. Please provide antibody species and target sequence data. Bear in mind that catalog numbers may not be useful to those repeating similar work.
  3. Mineralized matrix production in fig 2G is not convincing. Matrix is extracellular material produced only by dense cultures. Similarly, I see no convincing evidence of mineralizd matrix in 7F. This is a common problem with in vitro bone formation papers, but showing bone as an extracellular material containing mineralized collagen I is often not done. Bone does not form without osteoblasts lining matrix formation. With the in vivo data, this is not seen as a major issue but the data should not be shown in this form.
  4. Noting that the in vitro effects are in the order of 10% should be used to indicate that the overall effects of this pathway are confirmed, but that the overall effect on bone and adipocyte formation are partial, rather than absolute. 
  5. Possibly I missed something, but data on the specificity of JQ-1 should be cited, and it would be valuable to have a molecular method to back up the inhibitor effect. At a minimum, more controls regarding the effects of JQ-1 on BRD4 would be helpful. 

Round 2

Reviewer 1 Report

Inclusion of all the positive controls to Fig 1 suggests caution when interpreting these results. Overall it seems like HDAC4 is at most a partial contributor to GC effects on BMDCs (HDAC4 RNAi doesn't prevent GC inhibition of mineralization, OCN mRNA, Runx2 mRNA, Runx2 acetylation, or adipogenesis, and HDAC4 overexpression doesn't decrease mineralization). Perhaps the HDAC4 data should be removed or included as supplementary?

For the proximity ligation assay, the authors have not addressed my concern - the overwhelming presence of extranuclear H3K9ac signal suggests that the assay did not work. Since the assay did not work, it must be omitted from the paper.

In my previous review I asked the authors to interpret their finding that "loss of Foxp1 has no effect on mineralization and Runx2" and what that means in the context of either/or osteocyte adipocyte decision. In the response they describe the results but still do not interpret this finding. I would again ask for them to interpret these findings.

Author Response

Response to Reviewer #1
Inclusion of all the positive controls to Fig 1 suggests caution when interpreting these results. Overall it seems like HDAC4 is at most a partial contributor to GC effects on BMDCs (HDAC4 RNAi doesn't prevent GC inhibition of mineralization, OCN mRNA, Runx2 mRNA, Runx2 acetylation, or adipogenesis, and HDAC4 overexpression doesn't decrease mineralization). Perhaps the HDAC4 data should be removed or included as supplementary?
Authors’ response: We apologize for poor interpretation of analyses related to osteogenic differentiation of Hdac4 RNAi or cDNA-transfected cells. Investigations were that Hdac4 RNAi-transfected cells (column #3 in Fig. 1) displayed significant increases in mineralized matrix production, osteocalcin and Runx2 expression and H3K9ac occupation at Runx2 promoters than the scramble control group (column #1 in Fig. 1). Hdac4 RNAi-transfected cells also showed significant decreases in adipocyte formation and PPARγ2 expression, whereas H3K9ac binding PPARγ2 promoter was unaffected. Hdac4 knockdown significantly attenuated glucocorticoid-induced downregulation of osteogenic differentiation and H3K9ac binding Runx2 promoter. Hdac4 knockdown also significantly downregulated glucocorticoid-induced adipocyte formation. Forced Hdac4 expression (column #6 in Fig. 1) significantly decreased mineralized matrix production, osteogenic gene expression and H3K9ac enrichment in Runx2 promoters than the vector group. Hdac4 overexpression significantly enhanced adipocyte formation as compared to the vector group (column #5 in Fig. 1).

We re-arranged the figures and rewrote the sentences in the revised version. The revised text (Lines 210-225) now reads as follows:

Hdac4 knockdown increased H3K9ac levels (Fig. 1b) and attenuated glucocorticoid-induced loss of von Kossa-stained mineralized matrix formation 18 days after treatment (Fig. 1c). Loss of Hdac4 function significantly reversed glucocorticoid-mediated inhibition of osteocalcin and osteogenic transcription factor Runx2 expression (Fig. 1d) and H3K9ac binding Runx2 promoter 24 hours after treatment as evident from chromatin immunoprecipitation (ChIP)-PCR analysis (Fig. 1e). Osteogenic differentiation and H3K9ac binding Runx2 promoter were significantly increased above baseline in Hdac4 RNAi-transfected cells as compared to the scramble control group. Forced Hdac4 expression downregulated H3K9ac levels (Fig. 1b), mineralized matrix deposition (Fig. 1c), osteocalcin and Runx2 expression (Fig. 1d), as well as significantly decreased H3K9ac enrichment in Runx2 promoter (Fig. 1e) than the vector group. As cells grew in an adipogenic condition, glucocorticoid or Hdac4 overexpesson significantly increased adipocyte formation as evident from fluorescent Nile red staining 12 days after treatment (Fig. 1f), as well as upregulated adipogenic transcription factor PPARγ2 expression 24 hours after treatment (Fig. 1g). The glucocorticoid-induced upregulation of adipocytes and PPARγ2 expression were decreased in Hdac4 RNAi-transfected cells. Of note, glucocorticoid, gain or loss of Hdac4 function did not significantly change H3K9ac enrichment in PPARγ2 promoter 24 hours after treatment (Fig. 1h), indicating that an epigenetic reading pathway may regulate glucocorticoid-mediated PPARγ2 transcription.

For the proximity ligation assay, the authors have not addressed my concern - the overwhelming presence of extranuclear H3K9ac signal suggests that the assay did not work. Since the assay did not work, it must be omitted from the paper.
Authors’ response: Thank you for this suggestion. The images of protein ligation are removed from Fig 2in the revised version.

In my previous review I asked the authors to interpret their finding that "loss of Foxp1 has no effect on mineralization and Runx2" and what that means in the context of either/or osteocyte adipocyte decision. In the response they describe the results but still do not interpret this finding. I would again ask for them to interpret these findings.

Authors’ response: We very much appreciate the Reviewer’s constructive suggestion. We wrote the sentences in the revised version. The revised text (Lines 429-436) now reads as follows:

Foxp1 signaling appeared to, at least in part, regulate glucocorticoid-mediated PPARγ2 transcription as BRD4 or Foxp1 knockdown reduced Foxp1 occupation at PPARγ2 promoter, ameliorating glucocorticoid-induced oil droplet overproduction. Analyses were in agreement with a study showing that Foxp1 intensively influences PPARγ2 transcription to regulate adipocyte function [28]. This study revealed that Foxp1 was required to accelerate the shift of glucocorticoid-treated mesenchymal stem cells into adipocytes at the expense of osteoblast differentiation. The investigations of unaffected Foxp1 occupation at Runx2 promoter in glucocorticoid-treated or Foxp1 RNAi-transfected cells further explained the adipogenesis-promoting role Foxp1 signaling did play in mesenchymal stem cells.
